# Climate as a Predictive Factor for Invasion: Unravelling the Range Dynamics of *Carpomya vesuviana* Costa

**DOI:** 10.3390/insects15060374

**Published:** 2024-05-21

**Authors:** Chuangju Feng, Facheng Guo, Guizhen Gao

**Affiliations:** College of Forestry and Landscape Architecture, Xinjiang Agricultural University, Urumqi 830052, China; fcj2223022357@163.com (C.F.); ooooriiii@outlook.com (F.G.)

**Keywords:** climate change, agricultural pests, ensemble model, climate similarity

## Abstract

**Simple Summary:**

Invasive alien species, such as *Carpomya vesuviana* Costa (Diptera: Tephritidae), threaten global biodiversity, agriculture, and public health. This study used advanced modelling techniques to predict the potential distribution of *C. vesuviana* and understand its environmental impacts. The findings suggest that climate change is driving the geographic expansion of *C. vesuviana*, primarily in Asia, Africa, and Australia. The species is projected to shift poleward by the 2090s, as the species will have shifted to the polar regions under the influence of climate change. Effective management strategies are crucial to mitigate the environmental and health impacts of its expansion.

**Abstract:**

Invasive alien species (IAS) significantly affect global native biodiversity, agriculture, industry, and human health. *Carpomya vesuviana* Costa, 1854 (Diptera: Tephritidae), a significant global IAS, affects various date species, leading to substantial economic losses and adverse effects on human health and the environment. This study employed biomod2 ensemble models, multivariate environmental similarity surface and most dissimilar variable analyses, and ecological niche dynamics based on environmental and species data to predict the potential distribution of *C. vesuviana* and explore the environmental variables affecting observed patterns and impacts. Compared to native ranges, ecological niche shifts at invaded sites increased the invasion risk of *C. vesuviana* globally. The potential geographical distribution was primarily in Asia, Africa, and Australia, with a gradual increase in suitability with time and radiation levels. The potential geographic distribution centre of *C. vesuviana* is likely to shift poleward between the present and the 2090s. We also show that precipitation is a key factor influencing the likely future distribution of this species. In conclusion, climate change has facilitated the expansion of the geographic range and ecological niche of *C. vesuviana*, requiring effective transnational management strategies to mitigate its impacts on the natural environment and public health during the Anthropocene. This study aims to assess the potential threat of *C. vesuviana* to date palms globally through quantitative analytical methods. By modelling and analysing its potential geographic distribution, ecological niche, and environmental similarities, this paper predicts the pest’s dispersal potential and possible transfer trends in geographic centres of mass in order to provide prevention and control strategies for the global date palm industry.

## 1. Introduction

In the era of rapid global environmental change, invasive alien species (IAS) pose a significant threat to ecosystem stability and production in sectors, such as agriculture, forestry, animal husbandry, and fisheries, owing to their notable impacts on biodiversity, the economy, and society. This issue has garnered increasing attention from both the scientific community and the general public [1,2,3,4]. With the acceleration of global economic integration, the frequency of biological invasion events is expected to increase over the next 30 years [5]. Insects, a major category of invasive species, spread globally through transportation, trade, and tourism [6,7]. The rapid development of international trade, including e-commerce, new trade routes, and major infrastructure projects, has significantly increased the risk of biological invasions; however, current legislation and scientific tools are inadequate to fully address this growing threat, highlighting the need for a new approach to mitigate the risks associated with exotic species [8]. Understanding the factors contributing to the geographic spread of IAS is essential for developing effective mitigation strategies, including the size of an invasive species, which can greatly influence its ecological and economic impacts. Therefore, identifying the relative importance of different predictive variables in determining the potential range expansion of IAS is crucial [9].

Despite significant advancements in understanding the stages of biological invasion, the key factors influencing the range expansion of IAS, particularly those with profound ecological and economic repercussions, remain insufficiently explored [1,8]. Among the many IAS, *Carpomya vesuviana* Costa is a notorious pest [10,11,12,13,14]. Originating in India, the distribution of this insect has widened owing to international trade, marking its presence in numerous countries, including Afghanistan, Armenia, Azerbaijan, Bosnia, Caucasus, China, Cyprus, Georgia, Iran, Italy, Mauritius, Oman, Pakistan, Russia, Tajikistan, Thailand, Turkmenistan, and Uzbekistan [15,16,17]. The larvae of *C. vesuviana* bore into date fruits, leading to yield losses exceeding 20% [18], which significantly affects international trade and the livelihoods of fruit farmers. Characterised by its high fertility, considerable generational overlap, rapid development, and small size, *C. vesuviana* poses a challenge for detection within the date trade and has the potential to cover a vast range once introduced [19,20,21,22].

The date palm fly infests all types of date palms, including *Z. jujuba*, *Z. jujubavar.* spinosa, *Z. muritania*, *Z. numularia*, *Z. lotus*, and others [23,24]. *C. vesuviana*, a significant pest of date palms, causes yield losses of 40–100% [11]. In its initial detection year in Turpan, Xinjiang, China, it affected nearly one-third of date palms [11]. Additionally, in Pakistan, it has damaged up to 45% of date palms in certain areas [12], while in India, infestation rates range from 12% to 78.5% in two regions [13] and have reached 73–100% in others [14].

Ecological niche models (ENMs) play a pivotal role in understanding and forecasting species distribution by integrating fundamental ecological principles, including species distribution equilibria, habitat saturation, niche conservatism, unrestricted dispersal abilities, and the minimal influence of biotic interactions on species distribution. These models facilitate the examination of interactions between species and their environments [23,24,25]. The application of ENMs in identifying potential habitats for invasive insects provides significant benefits, such as identifying areas for the introduction of specific biological predators as well as aiding in making management decisions. Recent studies have increasingly used ENMs to analyse invasive insect species. In recent years, Maxent, Biomod2, and Climex have become the most popular ecological niche models among researchers [26,27,28,29]. Furthermore, ensemble models (EMs) are increasingly used to enhance the accuracy of predictions by mitigating noise in individual species distribution models (SDMs). Here, EMs refer to a combination of multiple predictive models to produce a more accurate outcome, whereas SDMs are models specifically used to predict the geographic distribution of species. The use of EMs, which integrate various SDMs, has proven essential for generating more precise geographic distribution forecasts [30,31,32]. Indeed, previous studies have shown that EMs, by aggregating diverse predictions from different SDMs, tend to be more reliable than any single model for predicting species distributions [33,34,35].

To effectively predict the global potentially suitable areas and understand the niche dynamics of *C. vesuviana* under changing climatic conditions, we employed the biomod2 integration model. This approach allowed a comprehensive analysis of multivariate similarities and differences in environmental factors, aiming to enhance our understanding and management of *C. vesuviana* invasion. Our strategy centred on identifying key environmental factors to prioritise prevention and control in regions that share climatic similarities with invaded areas. Given the significant economic impact of *C. vesuviana* on date palms, this study focuses on exploring its potential distribution and ecological niche shifts on a global scale. We will conduct an in-depth analysis of its environmental adaptations and geographic distributional centre-of-mass shifts, aiming to shed light on the dynamics and environmental drivers of its invasion, thereby providing a scientific basis for the development of effective biosecurity measures and management strategies. This methodology involved (1) the selection of superior single models for integration within biomod2 followed by the identification of the most effective integrated models; (2) using the optimal integrated model to forecast suitable areas for *C. vesuviana* under the SSP126 and SSP585 climate scenarios for 2050 and 2090; (3) examining the niche dynamics for *C. vesuviana* and comparing its invasion zones with its native regions; and (4) analysing environmental similarities within identified suitable areas based on a multivariate environmental similarity surface (MESS) and difference analyses to predict future primary spread zones and shifts in the dominant environmental factors. This structured approach was designed to offer insights into the adaptive strategies of *C. vesuviana*, thereby facilitating targeted and effective management practices aimed at mitigating the impacts of this species on global agriculture and biodiversity.

## 2. Materials and Methods

### 2.1. Species Geographical Distribution Data

We sourced the occurrence records of *C. vesuviana* from the Global Biodiversity Information Facility, the relevant literature [36,37,38,39,40,41,42,43,44,45,46,47,48,49], and various reports issued by government departments. In addition, we supplemented our dataset with records from field surveys conducted in China between 2021 and 2024 that specifically targeted *C. vesuviana*. To ensure compatibility with the resolution of our environmental variables, we employed ENM tools to refine our occurrence data [50]. This involved filtering the data to maintain only one distribution point per 5 × 5 km grid, effectively minimising spatial autocorrelation. If not addressed, spatial autocorrelation can lead to skewed results because of the clustering of data points [51,52]. After this filtering process, we compiled a set of 94 valid distribution points for *C. vesuviana* (Figure 1, Appendix A). This dataset formed the foundation for our predictive modelling, ensuring a robust analysis of species distribution in relation to environmental factors.

### 2.2. Environmental Variables

Firstly, in this study, because dates are widely distributed globally as a cash crop, their growth and development are subjected to heavy anthropogenic interference and their habitats are not expected to change significantly under future climate change [53,54], and because *C. vesuviana* coincides with the ripening period of date fruits [13], *C. vesuviana’s* 2~3 generations per year correspond to early and late fruits [55], and the damage rate varies according to year and host; therefore, biological factors are not significant. Secondly, in previous studies by scholars, it was found that it is mainly air temperature, humidity, and soil moisture that affect *C. vesuviana*, resulting in changes in air temperature, humidity, and soil moisture that mainly depend on precipitation and temperature, while the rest of the soil factors have minimal impacts [53,54,56,57], which is why 19 bioclimatic factors were used in this study. Therefore, 19 bioclimatic factors were selected to model *C. vesuviana* in this study.

We focused on 19 bioclimatic variables (Table 1) spanning the current (1970–2000) and future time frames (2050s: 2041–2060, 2090s: 2081–2100). Climate data for both periods were sourced from the Worldclim2.1 database (http://www.worldclim.org/, accessed on 5 December 2023, ensuring the comprehensive coverage of bioclimatic factors from bio1 to bio19 for each period. For future climate projections, we selected data from the BCC-CSM2-MR model of the Coupled Model Intercomparison Project (CMIP6). This choice marks a departure from the representative concentration pathways (RCPs) utilised in CMIP5, which incorporate a combination of shared socioeconomic pathways (SSPs) with RCPs to reflect future socioeconomic development scenarios [58].

To analyse these bioclimatic factors, we employed Pearson correlation analysis, and the variance inflation factor (VIF) was used to evaluate their intercorrelations and significance [59]. Using the R programming language, we conducted Spearman’s correlation and multicollinear VIF assessments on the interpolated data from our occurrence points. Our criteria for selecting relevant factors included those with a correlation coefficient <0.8 and VIF values < 10 [60]. VIF, which serves as the inverse of tolerance, is a metric used to quantify the extent of multicollinearity among variables. Specifically, a VIF value below 10 indicates the absence of multicollinearity, values between 10 and 100 indicate multicollinearity, and values above 100 indicate severe multicollinearity between factors [61]. This methodology facilitates the identification of the most relevant and independent bioclimatic variables for our analysis, ensuring the robustness of our ecological niche model. The filtered modelling factors are shown in Table 1.

### 2.3. Accuracy Evaluation of Single and Ensemble Models

We aimed to model the current and future geographic distributions of *C. vesuviana* by leveraging global occurrence data and environmental variables using the biomod2 package in R Studio (version 0.1.0, 2014) [62]. By integrating multiple models, we can leverage their individual sensitivities and explanatory powers to enhance the diversity, robustness, and comprehensiveness of predictions. This method balances each model’s biases, reduces errors, and, by merging their outputs, it improves forecast accuracy and better assesses uncertainty [30]. Studies indicate that integrating multiple models generally results in higher prediction accuracy than using a single model, thus enhancing the reliability and practicality of scientific research [34]. So, within biomod2, we utilised a comprehensive array of the following 12 modelling algorithms: artificial neural networks (ANN), classification tree analysis (CTA), flexible discriminant analysis (FDA), generalised additive models (GAM), generalised boosting models (GBM), generalised linear models (GLM), multivariate adaptive regression splines (MARS), maximum entropy (MAXENT), MAXNET, a random forest (RF), XGBOOST, and a species range envelope (SRE).

To refine the models, a BioModel tuning command was employed to optimise the parameters. We designated 75% of the data as the training set, ensuring an equal weight age for both the distribution and pseudo-distribution data to maintain the model balance. The modelling process was iterated five times to enhance reliability, resulting in 60 simulation models. From these, single models achieving a true skill statistic (TSS) greater than 0.7 were selected for further consideration, indicating high predictive accuracy and reliability [63].

An EM was then constructed to simulate the potential distribution areas of *C. vesuviana* using several ensemble methods, including the EMmean (ensemble mean), EMcv (coefficient of variation), EMci (confidence interval), EMmedin (median), EMca (consensus average), and EMwmean (weighted mean). This multifaceted approach allowed for a comprehensive evaluation of the models, ensuring the selection of the best integrated model for the subsequent prediction and analysis of *C. vesuviana* potential distribution. This systematic process underscores our commitment to employ advanced modelling techniques to accurately predict the spread of invasive species under varying environmental scenarios.

### 2.4. Migration of Centres of Potential Geographical Distributions and Overlapping Distribution Areas

The optimal model determined by our evaluation was used to map the currently suitable distribution area of *C. vesuviana*. We utilised the “bm_Find OptimStat” function within the biomod2 framework to determine the presence/absence (0/1) of the cut-off value. This cut-off helped to differentiate between unsuitable and suitable areas, where values below the cut-off were deemed unsuitable and those above it were categorised into three equal segments representing low-, moderate-, and high-suitability areas [64].

ArcGIS (10.4) software was used for spatial analysis and visualisation. Within this platform, the binary species distribution modelling tool available in the SDM tools plugin facilitated the examination of changes in the geographic distribution centres of *C. vesuviana* [65]. This approach allowed us to quantitatively assess shifts in species distribution centres, thereby providing a clearer understanding of potential expansion or contraction in response to environmental changes. This methodological approach underscores the utility of integrating statistical modelling with geographic information systems (GIS) to enhance the predictive accuracy and spatial resolution of species distribution forecasts.

### 2.5. Quantification of the Ecological Niche

We performed a quantitative niche analysis of *C. vesuviana* under diverse temporal and climatic conditions, leveraging environmental and distribution data specific to various periods and climates. For the niche analysis and visualisation, we employed the “ecospat” package within R [66], which is a comprehensive tool for ecological niche modelling and species distribution analysis. This package facilitates the calculation of the niche overlap index (D), which quantifies the extent of niche overlap between different periods or conditions on a scale from 0 (no overlap) to 1 (complete overlap) [67]. This index is crucial to understand how the ecological niche of *C. vesuviana* shifts or remains consistent across different climatic scenarios.

Additionally, the niche breadth module within ENMTools was utilised to calculate the niche width of *C. vesuviana* in both the current and projected future scenarios. In this context, B1 represented the minimum niche width and B2 denoted the maximum niche width, which were derived from the potential distribution data of the species. The niche width provides insights into the ecological versatility of *C. vesuviana*, indicating its ability to thrive across a range of environmental conditions. Thus, our analysis offers valuable perspectives on the adaptability and potential spread of *C. vesuviana* in response to climate change and contributes to the development of informed conservation and management strategies.

### 2.6. Analysis of MESS and Most Dissimilar Variable (MoD)

Environmental variables from the current potential distribution of *C. vesuviana* served as reference points for evaluating climatic anomalies in suitable habitats under various climate scenarios, both past and future. A MESS was employed to assess these anomalies, pinpointing the variable with the greatest dissimilarity, to identify the key factors influencing changes in the potential distribution of *C. vesuviana*.

The MESS calculation involved comparing the predictive variables (V1, V2, Vi, etc.) with reference conditions. Within this framework, “mini” and “maxi” represent the minimum and maximum values of the environmental variable Vi, respectively. The value of Vi at a specific point, P, during a given period is denoted by “pi”, and “fi” signifies the percentage of points in the study area where Vi is less than “pi” [68]. The calculation of the MESS values was based on the following criteria: (1) when fi = 0, the MESS was calculated as 100 × (p − mini)/(maxi − mini), indicating the lower limit of environmental suitability; (2) for values of fi between 0 and 50, the MESS was 2 fi, suggesting a gradient of increasing similarity; (3) for fi values greater than 50 and less than 100, the MESS was 2 × (100 − fi), reflecting a gradient of decreasing similarity; and (4) when fi = 100, the formula was modified to 100 × (maxi − pi)/(maxi − mini), indicating the upper limit of environmental suitability [69,70,71].

A negative MESS value indicated that the environmental conditions at point P exceeded the established limits of the reference environment for the species, marking it as an anomaly. Conversely, a value of 100 indicated complete agreement with the reference environmental conditions, suggesting no climatic anomalies. This analysis aimed to identify the most dissimilar variable at any given point, which could be crucial in understanding and predicting a species’ geographic shifts in response to climate change. To further refine the study, the MaxEnt model was enhanced with the density.tool.novel tool [72], integrating these environmental assessments to predict changes in the distribution of *C. vesuviana* accurately. This approach provides a nuanced understanding of the potential distribution shifts of the target species, focusing on identifying and addressing climatic anomalies that may affect habitat suitability.

## 3. Results

### 3.1. Model Accuracy Evaluation

To rigorously assess the model accuracy for predicting the distribution of *C. vesuviana*, we evaluated a comprehensive set of algorithms (ANN, CTA, FDA, GAM, GBM, GLM, MARS, MAXENT, RF, XGBOOST, SRE, and the EM). The performance metrics used were the TSS and area under the receiver operating characteristic curve (AUC). Among these models, ANN, GAM, GLM, MARS, MAXENT, MAXNET, RF, and XGBOOST demonstrated a superior performance, with TSS values exceeding 0.75 and AUC values surpassing 0.9 (Figure 2). Based on these results, eight models were selected to construct the EM. The EMca version of the EM exhibited the highest accuracy, with a TSS greater than 0.85 and an AUC exceeding 0.98 (Figure 2). Given its outstanding performance, the EMca model was selected for further visualisation and analysis.

### 3.2. Current Potential Global Geographic Distribution of C. vesuviana

The EMca was used to predict the current potential geographic distribution of *C. vesuviana* from 1970 to 2013. This prediction was visualised using ArcGIS software (version 10.4), resulting in a detailed distribution map under the prevailing climatic conditions (Figure 3). According to this map, the distribution of *C. vesuviana* is predominantly found across the following global regions: (1) Central Asia, encompassing the entirety of India, the India–Pakistan border, parts of southern and eastern Iran, the Afghanistan–Tajikistan border, and Xinjiang in China; (2) northern Australia, central regions of Africa (spanning latitudes −15.308 to −31.488 and longitudes 15.794 to −4.452); (3) South America, particularly Brazil, Chile, and Argentina; (4) North America, with occurrences in the western United States and central Mexico; and (5) Europe, along the border between Russia and Mongolia.

Our analysis indicated that the highly suitable habitats for *C. vesuviana* were primarily located in India and Iran, with moderately suitable areas extending from these central points. The simulation and prediction outcomes aligned closely with the known geographic distribution data for *C. vesuviana*, confirming the accuracy of the model. Furthermore, by analysing the number of grid cells corresponding to suitable habitats, we quantified the total area encompassed by these zones. As shown in Table 2, under current climatic conditions, suitable habitats for *C. vesuviana* covered an area of approximately 2383.75 × 10^4^ km^2^, which constitutes approximately 15.99% of the total global land area. The area categorised as highly suitable comprised approximately 283.72 × 10^4^ km^2^, accounting for 1.9% of the global land surface. This extensive coverage suggests that *C. vesuviana* occupies a significant portion of habitable zones in various selected regions, with highly suitable areas being more localised. These findings underscore the adaptability and potential spread of *C. vesuviana* across diverse environments, highlighting the importance of monitoring and managing its presence to mitigate its ecological and economic impacts.

### 3.3. Ecological Niche Analysis

The ecological niche space of *C. vesuviana* was analysed and visualised using various climate models, revealing insights into its adaptability and potential future distribution patterns. Figure 4 illustrates that *C. vesuviana* is expected to experience minimal loss in ecological niches in the future, with an expansion of ecological niches correlating with time and radiation intensity.

We also quantified the ecological niche overlap values of *C. vesuviana* across different climate scenarios, revealing significant ecological niche differentiation. The maximal ecological niche overlap (D50126) was 0.77, indicating substantial similarity under certain conditions, whereas the minimal ecological niche overlap (D90126) decreased to 0.53, indicating considerable differentiation in other climatic contexts. This suggests that *C. vesuviana* has broad ecological adaptability to various environments. Moreover, the analysis showed a decreasing trend in ecotope overlap values with an increasing radiation intensity over time. Specifically, the ecotope overlap values for the climate scenarios projected for 2090 were markedly lower than those for 2050. This trend suggests an expanding area invaded by *C. vesuviana* in the future, likely because of its ability to exploit a wider range of habitats as environmental conditions change.

Using the ENMtools software package, version 1.0, the ecological niche width of *C. vesuviana* was calculated, showing a maximum B1 value of 0.4808 and a minimum of 0.3735, whereas the maximum B2 value was 0.9705 with a minimum of 0.9601 (Table 3). The minimal variation between B1 and B2 across different periods indicated that *C. vesuviana* tended to exhibit generalist traits, preferring a wide array of habitats and resources. Both B1 and B2 values were observed to increase under future climate scenarios compared with the current period. This increase suggests that *C. vesuviana* is likely to occupy a larger ecological niche in the future, benefiting from the enhanced availability and distribution of resources as well as its intrinsic adaptability to changing environmental conditions. This adaptability positions *C. vesuviana* as a species capable of expanding its range in response to climate change, necessitating ongoing monitoring and management strategies to mitigate its potential impacts on ecosystems and human activities.

### 3.4. Future Potential Global Geographic Distribution of C. vesuviana

Using ArcGIS software, the EMca model predictions from biomod2 were reclassified to construct attribute lists and map the future suitable areas for *C. vesuviana* (Figure 5). This process facilitated the calculation of suitable habitat areas under various future climate scenarios, as detailed in Table 4. The analysis indicates a substantial increase in the suitable area for *C. vesuviana* compared to the current climate scenario, with the extent of this increase positively correlated with both the passage of time and rising levels of radiation. The smallest increase, 9.35%, is projected under the 2050s SSP126 scenario, whereas the most significant expansion is expected under the 2090s SSP585 scenario, which is described as exponential.

In the future, the habitat of *C. vesuviana* will continue to span key regions, including Afghanistan, Argentina, Brazil, Chile, China, India, Iran, Mexico, Pakistan, Tajikistan, the United States, and various African countries. Notably, primary changes are expected in areas currently classified as having low-to-medium habitability. These areas are anticipated to transform into high-habitability zones, with the current high-habitability zones expanding into surrounding areas. These projections suggest a rapid expansion of *C. vesuviana* into high-habitability zones in the future, potentially leading to a decrease in local biodiversity. The transformation and expansion of suitable habitats highlights the adaptability of *C. vesuviana* to changing climatic conditions, underscoring the need for ongoing monitoring and proactive management strategies to mitigate its ecological impacts.

### 3.5. Changes in Spatial Patterns of C. vesuviana

As shown in Figure 6 and Table 5, the expansion area increases gradually with the age and radiation level. In 2050, under the SSP126 scenario, the loss area will reach a maximum of about 72.87 × 10^4^ km^2^, mainly distributed in Asia, Oceania, Africa; under the SSP585 scenario, the expansion area will increase significantly to about 1396.02 × 10^4^ km^2^, the loss area will decrease to about 24.65 × 10^4^ km^2^, the expansion area will be distributed in North America, Asia, Europe, Oceania, and Africa, mainly concentrated in Asia and North America. In 2090, under the SSP126 scenario, the expansion area will increase with the age and radiation degree, mainly in Asia and North America; in 2090, under the SSP126 scenario, the loss area will reach a minimum of about 5.78 × 10^4^ km^2^, with only a small amount of loss in central Africa, and the expansion area will be further increased by about 2413.06 × 10^4^ km^2^, mainly in Asia and North America, and under the SSP585 scenario, the expansion area will reach a maximum of about 2956.58 × 10^4^ km^2^, but the loss area will remain small, with significant expansion in Africa and Asia.

### 3.6. MESS and MoD Variable Analysis

In the analysis of future climate scenarios for *C. vesuviana*, specifically the 2050s_SSP126, 2050s_SSP585, 2090s_SSP126, and the 2090s_SSP585 scenarios, the mean multivariate similarity values in comparison with the current (modern) distribution sites of *C. vesuviana* were calculated to be 16.66, 16.65, 16.60, and 13.21, respectively (Figure 7 and Figure 8). The scenario for 2050s_SSP126 exhibited the highest climatic similarity and lowest anomaly among the projections, indicating a close match to the current climatic conditions. Conversely, the 2090s_SSP585 scenario showed the lowest climatic similarity and the highest anomaly, suggesting significant deviations from current climate conditions and, therefore, a higher degree of climatic novelty for the potential distribution of *C. vesuviana*.

Our analysis identified the least similar variables affecting the modern suitable habitat of *C. vesuviana* as the precipitation of the coldest quarter, precipitation of the warmest quarter, mean diurnal range, and precipitation seasonality (Figure 8). Among these, precipitation emerged as the predominant element within the acclimatised area, indicating that changes in precipitation patterns might play a critical role in defining the future distribution of *C. vesuviana*. Temperature factors accounted for a small proportion of the acclimatised areas, suggesting that precipitation changes could have a more significant impact on species distribution than temperature changes.

This analysis underscores the importance of considering specific climatic variables, such as precipitation and temperature, in predicting the potential future distribution of species, such as *C. vesuviana*. Understanding these dynamics is crucial for developing effective conservation and management strategies to mitigate the potential impacts of shifts in the distribution of invasive species on biodiversity and ecosystem health under future climate change scenarios.

### 3.7. Environmental Factor Response Curves for C. vesuviana

The relationship between the presence probability of *C. vesuviana* and environmental factors is depicted by the response curves in Figure 9. A probability greater than 0.5 indicates that the environmental conditions are conducive to the species’ growth. According to Figure 9, the optimal mean diurnal temperature range for *C. vesuviana* is between 10.11 and 11.20 °C. Additionally, the species thrives with precipitation as follows: over 105.49 mm in the wettest month, 1.14 to 6.73 mm in the driest month, 63.45 to 166.81% seasonality, 22.52 to 791.88 mm in the warmest quarter, and over 1000 mm in the coldest quarter.

### 3.8. Centres of Potential Geographical Distributions of C. vesuviana

Figure 10 highlights significant shifts in the centre of the potential geographical distribution of *C. vesuviana* under current and future climate scenarios, evidencing the species’ response to climatic changes. Initially, the distribution centre was located in Afdem, Shinile, Ethiopia (41.0094° E, 10.2807° N). The observed shifts under various climate scenarios are as follows: In the SSP126 scenario for the 2050s, the centre of mass moved to Hulet Ej Enese, East Gojam, Ethiopia (38.0519° E, 11.1449° N), indicating a northwest shift of approximately 336.89 km. In the SSP126 scenario for the 2090s, the centre of mass transitioned to Melut, Upper Nile, South Sudan (33.1700° E, 10.3856° N), marking a south-westerly movement of approximately 539.58 km. Under the more severe SSP585 climate scenario, by the 2050s, the centre of mass was found in Baliet, Upper Nile, South Sudan (32.1586° E, 9.8577° N), reflecting a significant south-westerly shift of 969.50 km. By the 2090s, the centre moved slightly to the northeast, settling in Abu Jubayhah, South Kurdufan, Sudan (32.4675° E, 11.0285° N), with a relocation distance of 134.40 km.

These shifts under the different SSP scenarios suggest a general migration trend of *C. vesuviana* towards higher latitudes and altitudes, albeit with varying distances and directions depending on the specific future climate conditions modelled. Such migration patterns imply adaptability to changing climatic conditions, potentially affecting local ecosystems, agriculture, and biodiversity conservation efforts in new regions. The variance in migration distances and directions under different scenarios underscore the complexity of predicting changes in the species distribution in response to global climate dynamics.

## 4. Discussion

### 4.1. Impact of Environmental and Distributional Data on the Performance of SDMs

The accuracy and reliability of SDMs were significantly influenced by climatic variables and the distribution point data used to inform them. A critical balance is required; too many distribution points can lead to model overfitting, whereas climatic data that are too similar can result in excessive covariance in model outcomes [73,74]. Our study underscores the importance of the completeness and selection of species distribution points for enhancing model accuracy, particularly in the context of screening climatic data for model inputs. Based on previous research, modelling efforts that incorporate a comprehensive yet carefully selected array of species distribution data can mitigate the risk of overfitting commonly encountered in SDMs [75,76].

To address the potential biases introduced by environmental variables and distribution data, this study focused on meticulously screening single models for integration, followed by a rigorous evaluation of the integrated models. This approach aimed to refine the predictive capabilities of the model by minimising the uncertainty associated with the input data.

In addition, the duration of residence since the invasion of an IAS influences its range expansion. Longer residence times are generally associated with an increased adaptability and expansion capacity [77,78]. *C. vesuviana*, which originates from India [9], exemplifies this trend, as we predicted a high-fitness zone for this pest across a substantial area of its native region. This prediction aligns with observations from previous studies, suggesting that residence time plays a crucial role in the spatial distribution and fitness of invasive species. The model’s projection of *C. vesuviana* dominance in India not only reflects its historical presence, but also indicates potential areas of concern for biodiversity and agricultural practices, emphasising the need for ongoing surveillance and management strategies.

### 4.2. Relationships between Environmental Variables and Changes in the Potential Geographical Distribution of C. vesuviana

The successful global invasion of *C. vesuviana* has been largely attributed to the synergistic effects of bioclimatic variables and anthropogenic factors [79]. The interactions between these factors are complex and multiplicative, contributing significantly to the invasive capabilities of the species. Our study identified precipitation-related variables, specifically the precipitation of the wettest month, precipitation of the driest month, and precipitation seasonality, as critical determinants of *C. vesuviana*’s potential geographic distribution. In addition, anthropogenic factors, particularly international trade, played a pivotal role in the spread of *C. vesuviana*. The transportation of pests through commodities, such as dates, and other human-mediated activities, such as soil movement, seed dispersal by mowing or agricultural machinery, and transport via vehicles and trains, have facilitated their widespread distribution [8,80]. These activities have enabled *C. vesuviana* to overcome geographical barriers, leading to its establishment across diverse ecological niches.

Favourable climatic conditions, particularly suitable precipitation patterns, are crucial for the survival and proliferation of *C. vesuviana* in newly invaded areas. The species’ lifecycle, particularly its pupation stage in the soil, is directly influenced by rainfall, with specific patterns during critical months significantly impacting adult emergence [55,81]. These findings highlight the significant role of precipitation in facilitating the invasion and establishment of this species in new territories. In previous studies on *C. vesuviana*, it was found that *C. vesuviana* nymphs and adults showed high survival rates and rapid developmental rates at a soil relative humidity ranging from 5% to 25% and air relative humidity ranging from 20% to 40% [12,82]. This indicates that *C. vesuviana* is adapted to arid environments and shows strong drought tolerance. In this study, the discussion mainly focused on precipitation and temperature, but the results also showed that *C. vesuviana* adapted to specific environmental conditions. In particular, *C. vesuviana* survived in wet months, with precipitation exceeding 105.49 mm, and in extreme conditions (e.g., precipitation reaching 22.52 to 791.88 mm in the warmest quarter and more than 1000 mm in the coldest quarter), showing adaptation to changes in humidity. The ability of *C. vesuviana* to survive and reproduce under more extreme environmental conditions indicates a high degree of environmental resilience.

The results of this study align with those of previous studies, affirming the importance of precipitation as a determinant of *C. vesuviana* distribution and invasive success [82]. Understanding these dynamics is essential for predicting future invasion patterns and developing effective management strategies to mitigate the effects of *C. vesuviana* on agriculture and biodiversity in invaded regions.

### 4.3. Ecological Niche Dynamics of C. vesuviana

Our analysis of the ecological niche heterogeneity of *C. vesuviana* across its native and invaded sites revealed a notable expansion of ecological niches in the invaded territories, with virtually no loss of ecological niches observed. This species occupies a significant portion of the climatic niche of its native range across various invaded sites. This observation aligns with previous findings, indicating that *C. vesuviana*’s habitat has expanded globally [83]. Notably, the broad parasitism of various date species has facilitated its ecological niche expansion in invaded areas. Factors, such as its high fecundity, generational overlap, rapid development, and small size, contribute to the elusiveness of this species in the date trade, making it difficult to detect and control, thereby allowing it to occupy extensive ranges post-invasion [19,20,21,22].

The successful invasion of new areas by *C. vesuviana*, which has led to the expansion of ecological niches in these regions, suggests a competitive advantage. The ongoing debate on ecological niche changes and conservatism among invasive alien plant species (IAPs) during invasion indicates varying outcomes. Although 65% of the 815 terrestrial IAPs studied exhibit significant ecological niche shifts [84], the majority of terrestrial IAPs in regions such as Eurasia, North America, and Australia show conserved ecological niches [85]. Our findings support the ecotope shift hypothesis, suggesting that despite ecological niche changes, *C. vesuviana* poses a continued risk of invasion and dispersal on a global scale. This underscores the importance of monitoring and managing this pest to mitigate its potential impacts on agriculture and biodiversity worldwide.

### 4.4. Applicability and Limitations of Model Predictions

Our findings align with previous studies that used single models, Climex and MaxEnt, to predict the geographic distribution of *C. vesuviana*. We expanded on this by employing 12 models and six integrated approaches, enhancing the accuracy and minimising the typical errors of single models. Ensemble models (EMs) outperform single models in accuracy when using identical data [86,87]. A key benefit of EMs is their ability to effectively merge multiple models, capturing the authentic ecological niche signal while reducing the “noise” from data errors and model uncertainties [29,30]. This advantage is particularly valuable for newly invasive species, whose ecological relationships are not yet well understood [88].

This study’s modelling and analysis process uncovered several significant limitations and challenges. Firstly, the incomplete nature of field surveys may lead to species distribution data that do not accurately represent actual conditions. This data bias is particularly pronounced in remote or complex terrains, where surveys tend to be less thorough compared to accessible areas. Secondly, omitting biological factors from the predictor variables means the model only simulates theoretical, not actual, ecological niches [89,90]. Additionally, the model’s limited applicability to future environmental conditions compromises prediction reliability [91]. Furthermore, unmodelled factors like the migration capacity, environmental barriers, and evolutionary responses to environmental changes also affect species distribution predictions [92]. Therefore, considering these factors’ potential impact is crucial when using these predictions for field surveys and conservation efforts.

## 5. Conclusions

The potential geographical distribution of *C. vesuviana* was significantly influenced by the synergistic effects of key precipitation variables, namely the precipitation of the wettest month, precipitation of the driest month, and precipitation seasonality. As such, global climate change is anticipated to play a pivotal role in altering the potential geographical distribution and regions of overlapping distributions of this species. Methodologically, the EM demonstrated superior accuracy over the other separate modelling approaches, underscoring the reliability of EM predictions regarding the potential distribution of *C. vesuviana*.

According to the EM forecasts, the primary regions susceptible to invasion by *C. vesuviana* include vast areas across Asia, Africa, and Australia. Throughout the invasion process, *C. vesuviana* has shown the ability to adapt and expand its ecological niche with minimal loss, indicating a high capacity for ecological adaptation and survival in diverse environments. Projected shifts in the climate under scenarios SSP126 and SSP585 for both the 2050s and the 2090s indicate a consistent expansion of suitable habitats for *C. vesuviana*. Additionally, the potential geographical distribution centre of *C. vesuviana* is expected to migrate towards higher latitudes over time, particularly by the 2090s. This highlights the adaptability of *C. vesuviana* to changing environmental conditions and underscores the need for vigilant monitoring.

Although this study has accurately predicted the potential distribution of *C. vesuviana* and its trends through integrated modelling, future research needs to explore in greater depth the mechanisms of adaptation to climate extremes, the effects of genetic diversity in population migration, and the specific impacts of invasions on native biodiversity and ecosystems. In addition, the study of its management and control strategies, as well as the development of optimised early detection and rapid response systems, will also be key research directions.

Given the expansive potential distribution and ecological flexibility of *C. vesuviana*, there is a pressing need for continuous surveillance to provide an early warning system against its further global spread during the Anthropocene. The implications of such invasions are profound, affecting biodiversity, agriculture, and ecosystem stability across invaded regions. Policymakers, researchers, and conservationists must collaborate to develop strategies to mitigate the risks posed by invasive alien plant species, such as *C. vesuviana,* to ensure the protection and preservation of global biodiversity.

## Figures and Tables

**Figure 1 insects-15-00374-f001:**
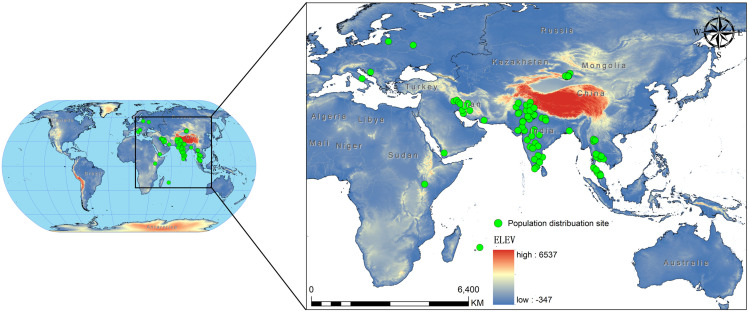
Study area geographic locations of *C. vesuviana* population distribution.

**Figure 2 insects-15-00374-f002:**
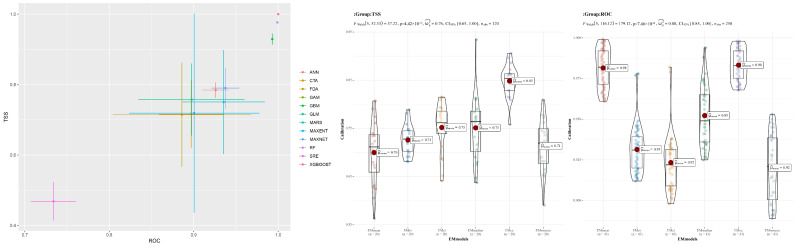
Single-model evaluation.

**Figure 3 insects-15-00374-f003:**
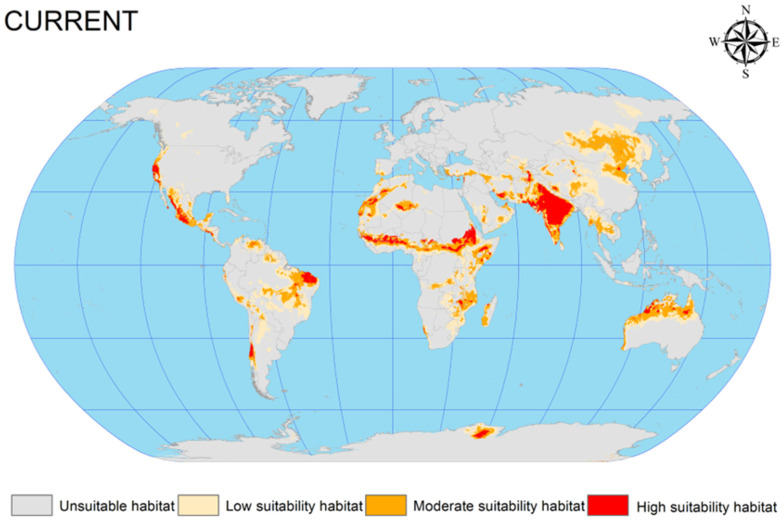
Current potential geographic distribution of *C. vesuviana*.

**Figure 4 insects-15-00374-f004:**
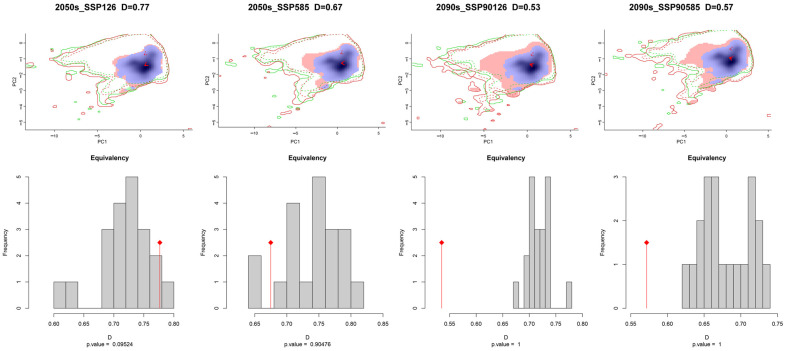
Niche differences of *C. vesuviana* in different climatic backgrounds in the future.

**Figure 5 insects-15-00374-f005:**
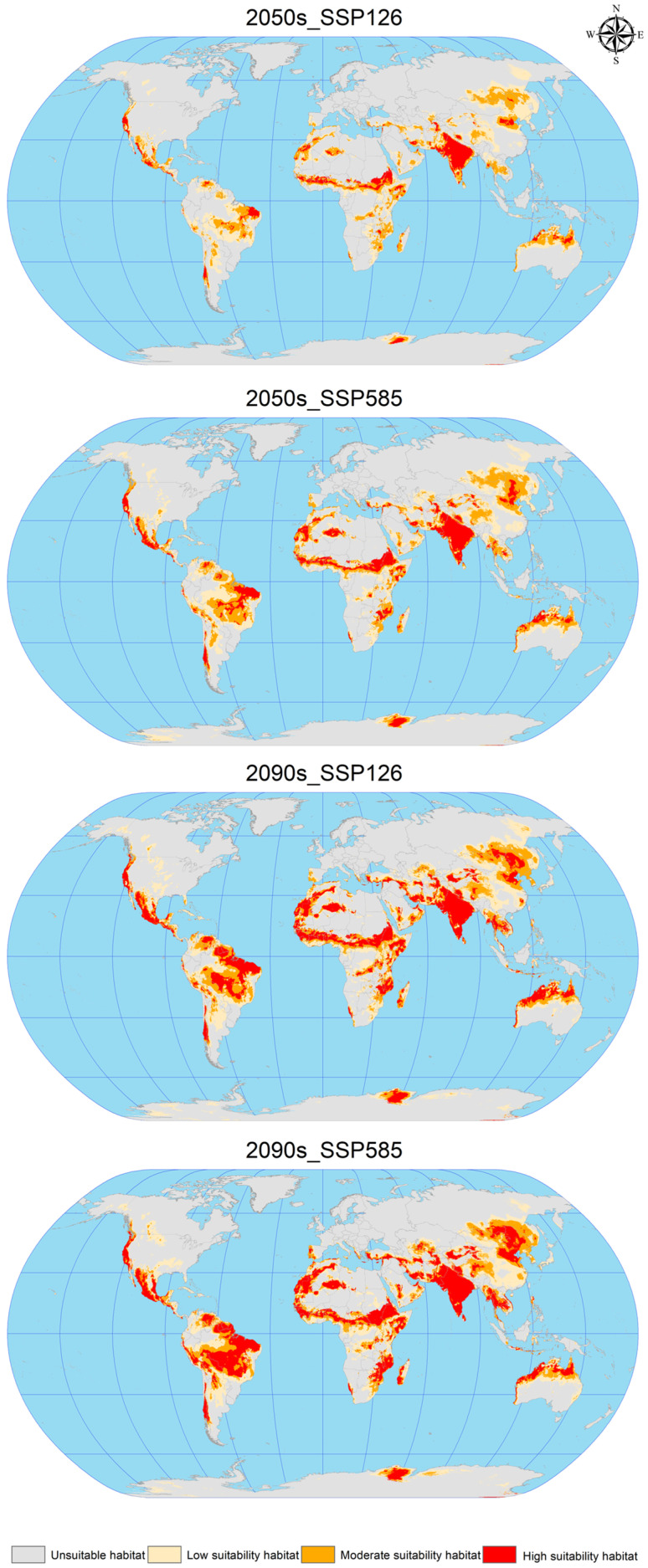
Geographic distribution of future scenarios for *C. vesuviana*.

**Figure 6 insects-15-00374-f006:**
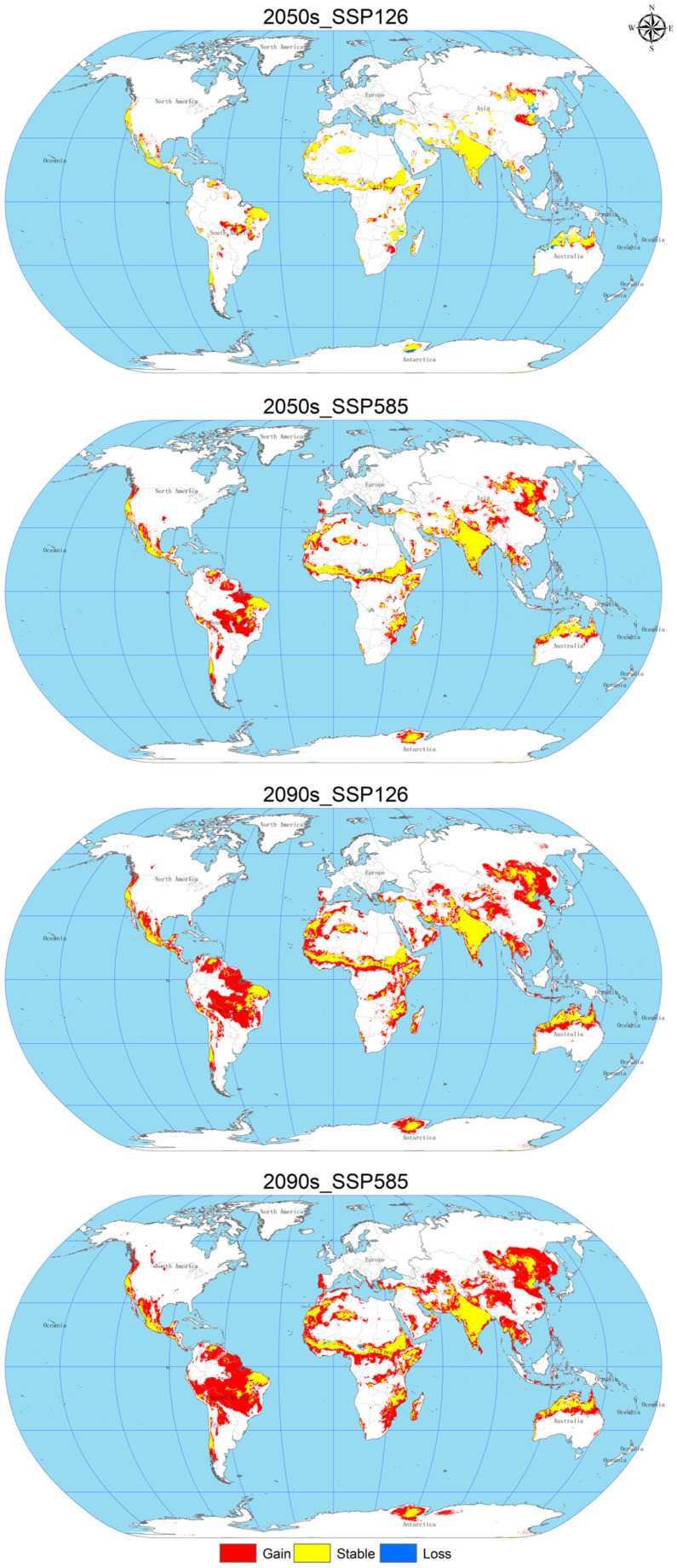
Changes in the spatial pattern of *C. vesuviana.*

**Figure 7 insects-15-00374-f007:**
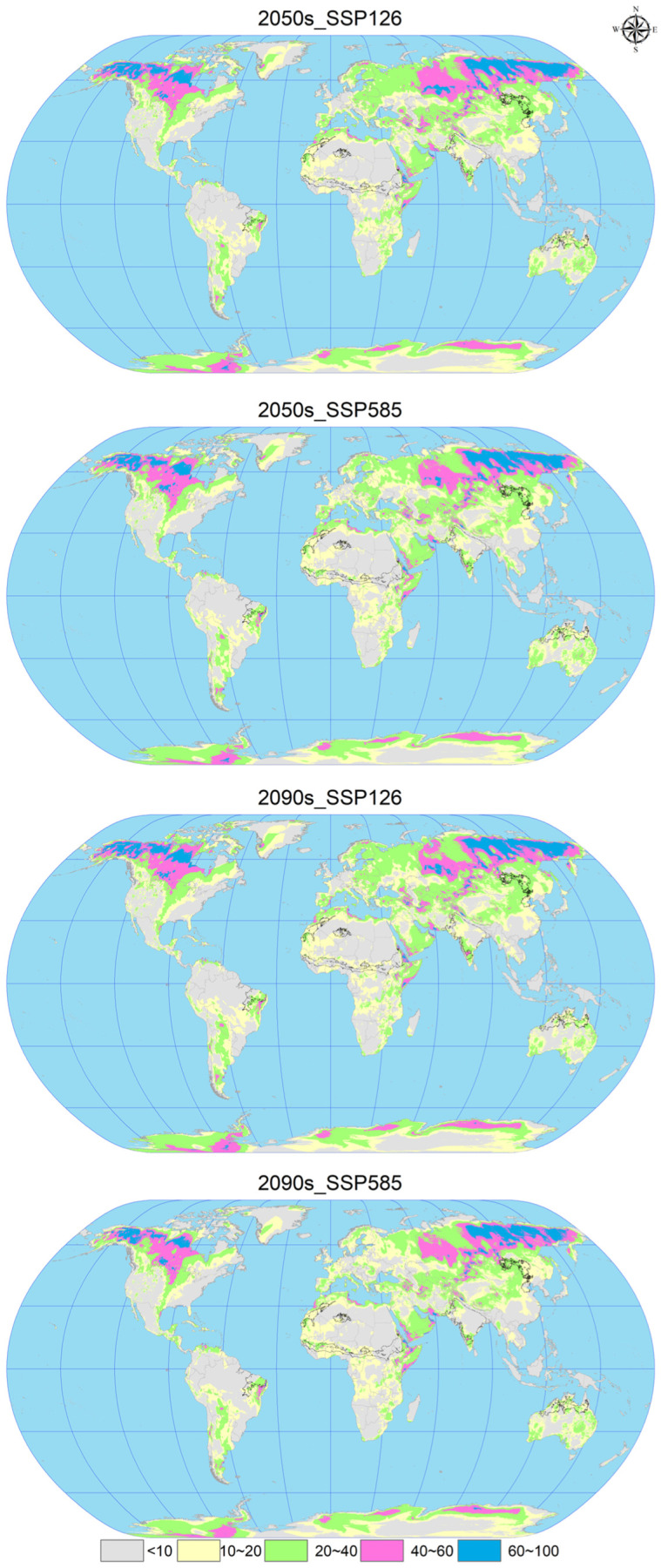
Multivariate environmental similarity surface (MESS) analysis for *C. vesuviana* under different combinations of climate change scenarios. The black line represents the outline of the modern habitable zone.

**Figure 8 insects-15-00374-f008:**
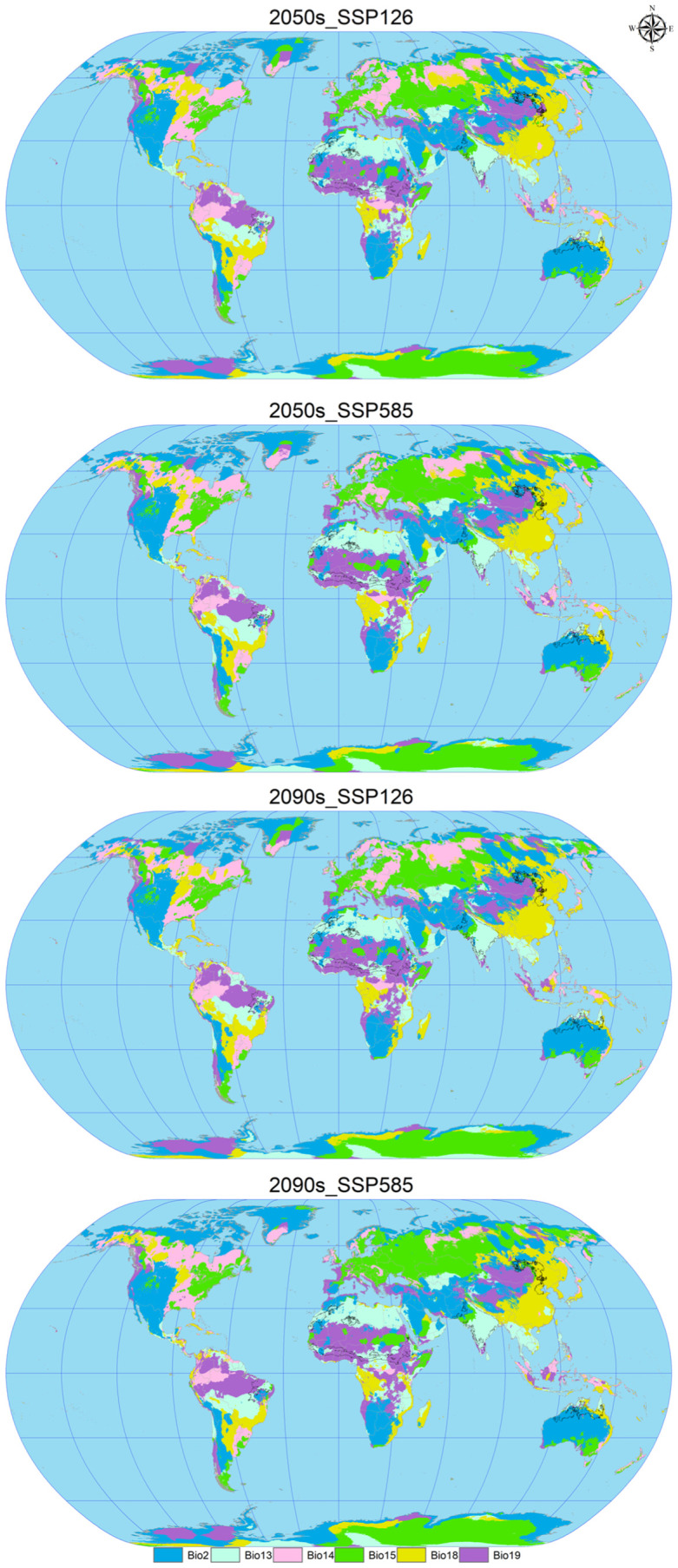
Multivariate least similar (MoD) variable analysis for *C. vesuviana* under different combinations of climate change scenarios. The black line represents the outline of the modern habitable zone.

**Figure 9 insects-15-00374-f009:**
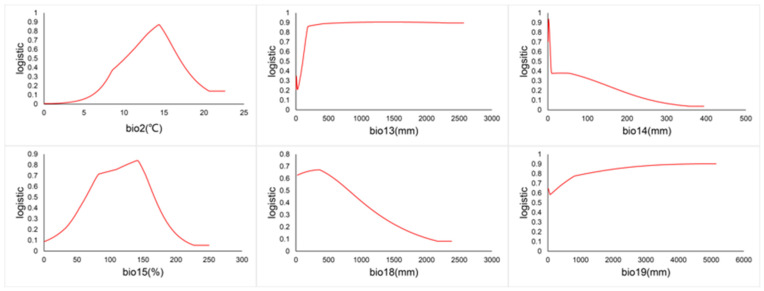
Environmental factor curves.

**Figure 10 insects-15-00374-f010:**
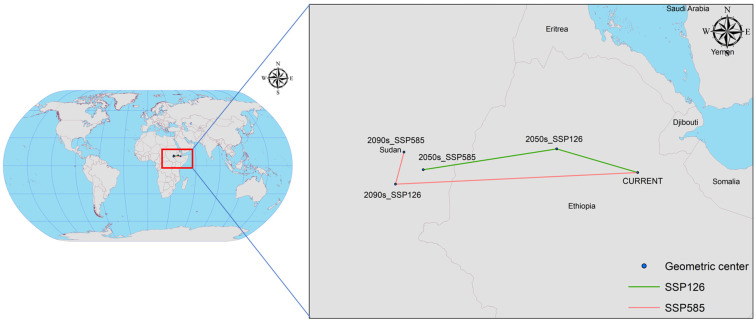
Centroid distributional shifts under different climate scenarios for *C. vesuviana*.

**Table 1 insects-15-00374-t001:** Three environmental factors.

Environmental Factors	Factor Name	Variable Name	Work Unit	VIF	Contribution	Permutation Importance
Bioclimatic factors	Bio2	Mean Diurnal Range	°C	3.8343	8.1146	1.4312
	Bio13	Precipitation of Wettest Month	mm	6.3589	27.4789	13.6867
	Bio14	Precipitation of Driest Month	mm	3.8343	7.4252	6.0334
	Bio15	Precipitation Seasonality	%	5.4600	27.1387	32.7995
	Bio18	Precipitation of Warmest Quarter	mm	3.3828	21.0308	34.1939
	Bio19	Precipitation of Coldest Quarter	mm	4.6173	8.8118	11.8552

**Table 2 insects-15-00374-t002:** Current global area (×10^4^ km^2^) of suitable habitat of *C. vesuviana.*

Habitat Type	Unsuitable	Low	Moderate	High
Area	12,516.24	1237.66	862.36	283.72
Percentage (%)	84.00	8.30	5.78	1.90

**Table 3 insects-15-00374-t003:** Ecological niche width of *C. vesuviana.*

Climate Scenarios	Niche Width	
	B1	B2
Proj50126	0.3814	0.9606
Proj50585	0.4353	0.9662
Proj90126	0.4808	0.9705
Proj90585	0.4612	0.9681
Projcurrent	0.3735	0.9601

**Table 4 insects-15-00374-t004:** Size of future suitable areas for *C. vesuviana* under different climatic conditions (×10^4^ km^2^).

Climate Scenarios	Unsuitable	Low	Moderate	High	ALL
2050s_SSP126	12,293.47	1336.58	898.67	371.27	2606.52
2050s_SSP585	11,477.49	1598.69	1143.23	680.58	3422.50
2090s_SSP126	10,802.94	1722.09	1204.86	1170.09	4097.51
2090s_SSP585	10,776.49	1642.67	1077.86	1402.96	4123.50

**Table 5 insects-15-00374-t005:** Area of change in spatial pattern of *C. vesuviana* (×10^4^ km^2^).

Climate Scenarios	Range Expansion	No Occupancy	No Change	Range Contraction	Gain (%)	Loss (%)
2050s_SSP126	369.85	13,294.76	1210.24	72.87	15.48	3.05
2050s_SSP585	1396.02	12,268.60	1258.45	24.65	58.58	1.03
2090s_SSP126	2413.06	11,251.55	1277.33	5.78	101.25	0.24
2090s_SSP585	2956.58	10,708.03	1270.28	12.8360	124.04	0.53

## Data Availability

The data presented in this study are available on request from the corresponding author.

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
