# Peer review of "Climate as a Predictive Factor for Invasion: Unravelling the Range Dynamics of Carpomya vesuviana Costa"

_insects, 2024, doi:10.3390/insects15060374_

Round 1

Reviewer 1 Report

Comments and Suggestions for Authors

The manuscript titled “Climate as a Predictor of Invasive Success: Unravelling the Range Dynamics of Carpomya vesuviana Costa predicted the potential distribution of Carpomya vesuviana and also explored the environmental variables affecting native ranges, global ecological niche shifts. The study succeeded in estimating  the potential global geographical distribution in terms of habitat suitability in future climatic conditions.  

Overall, this research is potentially full of interest, as it addresses the relevant topic to predict future habitat suitability of IAS. Moreover, there is still a lack of examples of this type of research. I believe that the study is drafted in a clear and intelligible fashion; the text is well organized.

I have few suggestions

Carpomya vesuviana is predominately known for its infection to most of the Ziziphus species in India as global as well. However, you have not mentioned in the introduction section about Ziziphus or other horticulture crops from the available literature. Insert a new paragraph describing the reported harmful effects of the model insect on regional and global scale.

In species geographical distribution data you have mentioned occurrence records from field surveys conducted in China between 2021 and 2024, please add information about the location and the host from which the occurrence points recorded.

In table 4, you should also mention the overall gain and loss in terms of percentage in all the future climatic scenarios for better readability and understanding.

Wish you luck

Author Response

Thanks very much for taking your time to review this manuscript. I really appreciate all your comments and suggestions! Please find my itemized responses in below and my corrections in the re-submitted files. Appended to this letter is our point-by-point response to the comments raised by the reviewers. The comments are reproduced and our responses are given directly afterward in a different color (red).

Thanks again.

Reviewer 2 Report

Comments and Suggestions for Authors

The research topic of the authors is relevant. The scope of work is large. However, there are some comments on the presented material.

The aim of the research must be declared at the end of the introduction and in the abstract.

When first mentioned, species names of insects must contain the name of the author and the year of description. The family must be specified. Carpomya vesuviana Costa, 1854 (Diptera: Tephritidae). In further mentions it is enough to write Cvesuviana (and without Costa!)

The Latin name of the host plant is necessary – Ziziphus jujuba (Ziziphus zizyphus) (Rhamnaceae)

lines 56–58 (and 333–334). Countries should be listed alphabetically or in the order of C. vesuviana invasion but indicate the years of invasion in parentheses in the second case.

line 136. Table 1. Three environmental factors. Why three? here are 19 bioclimatic factors which are used in MAxENt and other models. The 1st column “Bioclimatic factors” may be in the title of the table. “Work unit” may be in the same column as a variable name. But it is possible to add a column and to explain the sense of some variables (for example, as in this paper: O’Donnell, M.S.; Ignizio, D.A. Bioclimatic Predictors for Supporting Ecological Applications in the Conterminous United States; Data Series 691; U.S. Geological Survey: Reston, VA, USA, 2012. Available online: https://pubs.usgs.gov/ds/691/ds691.pdf (accessed on 10 August 2022).

General comment. In recent years, dozens of models and approaches to predicting the ranges of different species of insects and other animals and plants have been developed. Most models consider current climate parameters and their predicted values. However, when it comes to phytophagous insects, especially monophagous insects as the subject of this paper, it is imperative to consider the distribution and requirements of the host plant to environmental factors, primarily soil and climate. Under unfavorable conditions, the timing of the tree’s seasonal development may change, disrupting the synchronicity of the appearance of leaves or fruits necessary for the insect’s nutrition. The fly completes the full cycle in almost a month; will fruits be available for its nutrition after climate changes? The more generalized models are used, the less the environmental requirements of both the plant and the phytophagous insects are taken into account.

Therefore, here the main biological features of C. vesuviana, and the seasonal development of this fly and its host plant must be described.

Ziziphus jujuba is a preferable if not alone host plant for C. vesuviana. Using only meteorological variables may overestimate the area of this insect spread. It is necessary to indicate in what area Ziziphus jujuba is grown and whether an increase in this area is planned. Soil conditions and other limits of Ziziphus jujuba distribution must be considered too.

Author Response

Thanks very much for taking your time to review this manuscript. I really appreciate all your comments and suggestions! Please find my itemized responses in below and my corrections in the re-submitted files.Appended to this letter is our point-by-point response to the comments raised by the reviewers. The comments are reproduced and our responses are given directly afterward in a different color (red).

Thanks again!

Guizhen Gao

Reviewer 3 Report

Comments and Suggestions for Authors

Dear Authors

Regarding this study, I don't think there is much problem with the research methodology, but the writing needs to be strengthened, especially because there are a lot of subparagraphs in the text, which can actually be combined together. In addition, the presentation of images and important literature citations needs to be improved. Please refer to the file I provided

Comments on the Quality of English Language

I don't have direct advice on English writing, please contact a professional English editor.

Author Response

Thanks very much for taking your time to review this manuscript. I really appreciate all your comments and suggestions! Please find my itemized responses in below and my corrections in the re-submitted files.Appended to this letter is our point-by-point response to the comments raised by the reviewers. The comments are reproduced and our responses are given directly afterward in a different color (red).

Thanks again!

Round 2

Reviewer 1 Report

Comments and Suggestions for Authors

The manuscript has been revised and the authors significantly improved the manuscript. The revised version is suitable for publication. 

Reviewer 3 Report

Comments and Suggestions for Authors

Dear Authors

After considering the revised manuscript,

I think you have done a good job and met the acceptance criteria of the journal. I have only one small suggestion, in the "Simple Summary" section, you have mentioned the scientific name of the pest, including the author's name and the year of publication, if the year is included, the original descriptive article should be cited, so I suggest that you can omit the year.

Sincerely,